# How Context Affects Language Models' Factual Predictions

**Fabio Petroni**[1]                                    FABIOPETRONI@FB.COM
**Patrick Lewis**[1,2]                                        PLEWIS@FB.COM
**Aleksandra Piktus**[1]                                       PIKTUS@FB.COM
**Tim Rocktäschel**[1,2]                                        ROCKT@FB.COM
**Yuxiang Wu**[2]                                 YUXIANG.WU.18@UCL.AC.UK
**Alexander H. Miller**[1]                                         AHM@FB.COM
**Sebastian Riedel**[1,2]                                      SRIEDEL@FB.COM
[1]*Facebook AI Research*
[2]*University College London*

## Abstract

When pre-trained on large unsupervised textual corpora, language models are able to store and retrieve factual knowledge to some extent, making it possible to use them directly for zero-shot cloze-style question answering. However, storing factual knowledge in a fixed number of weights of a language model clearly has limitations. Previous approaches have successfully provided access to information outside the model weights using supervised architectures that combine an information retrieval system with a machine reading component. In this paper, we go a step further and integrate information from a retrieval system with a pre-trained language model in a purely unsupervised way. We report that augmenting pre-trained language models in this way dramatically improves performance and that the resulting system, despite being unsupervised, is competitive with a supervised machine reading baseline. Furthermore, processing query and context with different segment tokens allows BERT to utilize its Next Sentence Prediction pre-trained classifier to determine whether the context is relevant or not, substantially improving BERT's zero-shot cloze-style question-answering performance and making its predictions robust to noisy contexts.

## 1. Introduction

Pre-trained language models such as BERT [Devlin et al., 2019] and RoBERTa [Liu et al., 2019] enabled state-of-the-art in many downstream NLP tasks [Wang et al., 2018a, 2019, Wu et al., 2019]. These models are trained in an unsupervised way from large textual collection and recent work [Petroni et al., 2019, Jiang et al., 2019, Talmor et al., 2019, Devlin et al., 2019] has demonstrated that such language models can store factual knowledge to some extent. However, considering the millions of documents and facts in Wikipedia[1] and other textual resources, it unlikely that a language model with a fixed number of parameters is able to reliably store and retrieve factual knowledge with sufficient precision [Guu et al., 2020].

One way to get around this is to combine machine reading with an information retrieval (IR) system [Chen et al., 2017, Guu et al., 2020]. Given a question, the IR system retrieves

---

1. https://en.wikipedia.org/wiki/Wikipedia:Statistics

relevant contexts which are subsequently processed by a reading component. In the case of DrQA [Chen et al., 2017], the retriever is fixed and the reading component is trained based on retrieved contexts, whereas in REALM [Guu et al., 2020] the IR component is trained alongside the reader during both pre-training and subsequent fine-tuning.

In this paper, we go a step further and forego supervised fine-tuning. Instead, we consider the purely unsupervised case of augmenting a language model with retrieved contexts at test time. We demonstrate that augmenting pre-trained language models with such retrieved contexts dramatically improves unsupervised cloze-style question answering, reaching performance that is on par with the supervised DrQA approach. In addition to being unsupervised, using a pre-trained language model like BERT instead of a trained machine reading component has several other advantages. Since BERT is not an extractive QA model, it is able to utilize contexts that contain relevant information but do not contain the answer span directly. More importantly, we find that via the next-sentence prediction objective BERT is able to ignore noisy or irrelevant contexts.

In summary, we present the following core findings: i) augmenting queries with relevant contexts dramatically improves BERT and RoBERTa performance on the LAMA probe [Petroni et al., 2019], demonstrating unsupervised machine reading capabilities of pre-trained language models; ii) fetching contexts using an off-the-shelf information retrieval system is sufficient for BERT to match the performance of a supervised open-domain QA baseline; iii) BERT's next-sentence prediction pre-training strategy is a highly effective unsupervised mechanism in dealing with noisy and irrelevant contexts. The code and data to reproduce our analysis will be made publicly available.

## 2. Related Work

**Language Models and Probes** With the recent success of pre-trained language models like BERT [Devlin et al., 2019] and its variants [Liu et al., 2019, Seo et al., 2019, Raffel et al., 2019, Lewis et al., 2019a], it becomes increasingly important to understand what these models learn. A variety of "probes" have been developed to analyse the syntactic structures, such as syntax trees [Marvin and Linzen, 2018, Hewitt and Manning, 2019, Vig and Belinkov, 2019], negative polarity items [Warstadt and Bowman, 2019, Warstadt et al., 2019], semantic fragments [Richardson et al., 2019], function words [Kim et al., 2019], and many other linguistic phenomena [Tenney et al., 2019a,b, De Cao et al., 2020]. To measure the factual knowledge present in these pre-trained language models, Petroni et al. [2019] propose the LAMA benchmark which tests the models with cloze-style questions constructed from knowledge triples. Jiang et al. [2019] later extends LAMA by automatically discovering better prompts, Kassner and Schütze [2019] add negated statements, Poerner et al. [2019] filter out easy-to-guess queries, and Richardson and Sabharwal [2019], Talmor et al. [2019], Bisk et al. [2019] develop further probes for textual reasoning. Pre-trained language models have also been fine-tuned and adapted to be used as information retrieval systems [Yang et al., 2019b, Yilmaz et al., 2019, Seo et al., 2019].

**Open-Domain QA** Open-domain QA aims at answering questions without explicitly knowing which documents contain relevant information. Open-domain QA models often involve a retriever to find relevant documents given a question, and a reader to produce the answers [Chen et al., 2017]. Works in this areas mostly focus on enhancing retrieval

quality [Choi et al., 2017, Wang et al., 2018b,c, Lin et al., 2018, Min et al., 2018, Lee et al., 2018, 2019, Das et al., 2019, Xiong et al., 2019], improving answer aggregation [Clark and Gardner, 2018, Wang et al., 2018c, Lee et al., 2018, Pang et al., 2019], and accelerating the whole pipeline [Seo et al., 2019]. Recently, Guu et al. [2020] show that augmenting language model pre-training with a knowledge retriever induces performance gains on open-domain QA tasks. Our work differs from previous works in open-domain QA in two ways: i) we consider a fully unsupervised setting using a pre-trained language model and an off-the-shelve information retrieval system, ii) our aim is to assess the prediction of factual knowledge in this setup rather than to improve open-domain question answering in general.

## 3. Methodology

Given a cloze-style question $q$ with an answer $a$, we assess how the predictions from a language model change when we augment the input with contexts $c$. In this section, we describe the datasets we use to source $(q, a)$ pairs, as well as various methods of generating context documents $c$.

### 3.1 Datasets

We use the LAMA[2] probe in our experiments [Petroni et al., 2019], a collection of cloze-style questions about real world relational facts with a single token answer. Each question is accompanied by snippets of text from Wikipedia that are likely to express the corresponding fact. Although there are several cloze-style QA datasets (some listed in Section 2) we decided to use LAMA because: (1) the nature of the LAMA data is aligned with the relational knowledge focus or our analysis (*i.e.*, given a subject and a relation predict the object) and (2) each data point is aligned by construction with relevent contextual information. We consider the Google-RE[3] (3 relations, 5527 facts), T-REx [Elsahar et al., 2018] (41 relations, 34039 facts) and

| Corpus | Relation | Statistics | |
|---|---|---|---|
| | | #Facts | #Rel |
| Google-RE | `birth-place` | 2937 | 1 |
| | `birth-date` | 1825 | 1 |
| | `death-place` | 765 | 1 |
| | Total | 5527 | 3 |
| T-REx | 1-1 | 937 | 2 |
| | $N$-1 | 20006 | 23 |
| | $N$-$M$ | 13096 | 16 |
| | Total | 34039 | 41 |
| SQuAD | Total | 305 | - |

Table 1: Statistics for the LAMA data.

SQuAD [Rajpurkar et al., 2016] (305 questions manually translated in cloze-style format) subsets of the probe. More detailed statistics for the LAMA data considered are reported in Table 1. For the RoBERTa results, we trim the LAMA dataset (by about 15%) such that all answers are in the model's vocabulary, so BERT and RoBERTa numbers in this paper should not be directly compared as they consider slightly different subsets of the data.

---

2. https://github.com/facebookresearch/LAMA
3. https://code.google.com/archive/p/relation-extraction-corpus

## 3.2 Baselines

We consider DrQA [Chen et al., 2017], a popular system for open-domain question answering. The overall pipeline consists of two phases: (1) a TF-IDF document retrieval step, where the model finds relevant paragraphs from Wikipedia and (2) a machine comprehension step to extract the answer from those paragraphs. The machine comprehension component is trained with supervision on SQuAD v1.1 [Rajpurkar et al., 2016]. In order to apply DrQA to the LAMA probe, we take inspiration from [Levy et al., 2017] and map each cloze-style template to a natural question template (*e.g.*, "X was born in [Mask]" to "Where was X born?"). We constrain the predictions of DrQA to single-token answers as in Petroni et al. [2019]. Our results for DrQA and BERT are directly comparable with the other baselines in Petroni et al. [2019].

## 3.3 Language Models

Among the constellation of language models that have been proposed in recent years we consider BERT [Devlin et al., 2019] since it is one of the post popular and widely used at the time of writing.[4] Moreover, the large cased version of the BERT model is the best performing LM on the LAMA probe among those considered in Petroni et al. [2019]. We additionally consider the large version of the RoBERTa model [Liu et al., 2019]. Both BERT and RoBERTa have been trained on corpora that include Wikipedia. While BERT uses two pre-training strategies, Mask Language Modelling (MLM) and Next Sentence Prediction (NSP), RoBERTa considers only the MLM task. We produce a probability distribution over the unified vocabulary of Petroni et al. [2019] for the masked token in each cloze-style questions and report the average precision at 1 (P@1).

## 3.4 Contexts

We enrich cloze statements with different types of contextual information. We explicitly distinguish cloze question $q$ and context $c$ in the input according to the model. For BERT, we use different segment embeddings, index 0 for $q$ and 1 for $c$, and insert the separator token (*i.e.*, [SEP]) in between. For RoBERTa, which is not equipped with segment embeddings, we use the end of sentence (eos) token to separate $q$ and $c$. We addidionally performed some experiments without this clear separation of query and context, but considering them as concatenated in a single segment (or wihtout the eos token in between). The input is truncated to 512 tokens.

### 3.4.1 Oracle Contexts

We provide an oracle-based (ora) context in order to assess the capability of LMs to exploit context that we know is relevant to the entity in the question. Concretely, we use the Wikipedia snippet accompanying each example in the LAMA probe, truncated to at most five sentences. This context often contains the true answer and always contains related true information.

---

4. https://huggingface.co/models

| LAMA | Relation | B | B-ADV | *open domain sourced context* | | | B-ORA |
| | | | | B-GEN | DrQA | B-RET | |
|------|----------|-----|-------|-------|------|-------|-------|
| Google-RE | birth-place | 16.1 | 14.5 | 8.5 | **48.6** | 43.5 | *70.6* |
| | birth-date | 1.4 | 1.4 | 1.4 | 42.9 | **43.1** | *98.1* |
| | death-place | 14.0 | 12.6 | 6.0 | **38.4** | 35.8 | *65.1* |
| | Total | 10.5 | 9.5 | 5.3 | **43.3** | 40.8 | *78.0* |
| T-REx | 1-1 | 74.5 | 74.5 | 71.3 | 55.2 | **81.2** | *91.1* |
| | *N*-1 | 34.2 | 33.8 | 32.7 | 30.4 | **47.5** | *67.3* |
| | *N*-*M* | 24.3 | 23.6 | 23.8 | 15.4 | **32.0** | *52.4* |
| | Total | 32.3 | 31.8 | 31.1 | 25.8 | **43.1** | *62.6* |
| SQuAD | | 17.4 | 17.4 | 15.8 | **37.5** | 34.3 | *61.7* |
| *weighted average* | | 30.5 | 30.0 | 29.0 | 27.2 | **42.8** | *63.6* |

Table 2: Mean precision at one (P@1) for the DrQA baseline, BERT-large on context-free cloze questions (B) and on adversarial (B-ADV), generated (B-GEN), retrieved (B-RET) and oracle (B-ORA) context-enriched questions on the relational LAMA probe. The fully unsupervised B-RET is competitive with the supervised DrQA system and is dramatically better than the context-free baseline. We weight the average per number of relations (3 for Google-RE, 41 for T-REx and we consider SQuAD as a single contribution). Pairwise sign tests per relation show statistically significant differences (p-value below 1e-5) between: B-RET and all other results; B-ORA and all other results.

### 3.4.2 Sourcing Relevant Contexts

Relevant context is often not available and must be automatically sourced by the model [Chen et al., 2017, Clark and Gardner, 2018]. In this scenario, we consider two possible approaches: using an information retrieval engine (RET) or generating the context with an autoregressive LM (GEN) [Radford et al., 2019]. For the retrieval case, we use the first paragraph from DrQA's retrieval system as context. For the generative case, taking inspiration from the study of Massarelli et al. [2019], we consider a 1.4B parameters autoregressive language model trained on CC-NEWS [Liu et al., 2019]. This model has been shown to generate more factual text with respect to others trained on different corpora, including Wikipedia. For each question in LAMA, we use the natural question template as prefix to condition the generation, and generate five sentences using the delayed beam search strategy [Massarelli et al., 2019]. These results may be quite related to the entity in the query, though they may not always be completely factual.

### 3.4.3 Adversarial Contexts

We provide an uninformative context in order to test the ability of the model to ignore irrelevant context that is not useful for answering the query. We do this by randomly sampling an oracle context from a different question that has the same relation type but a different answer $a'$. This results in a context document that refers to a different subject

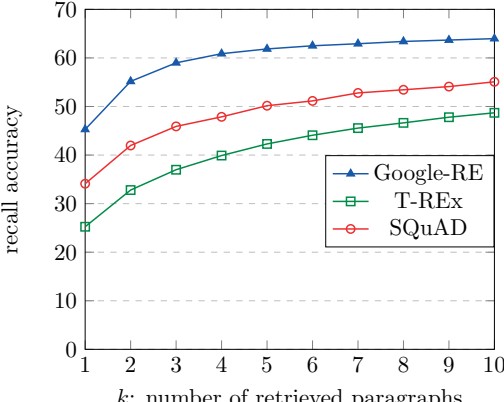

| P@1 | answer in ctx | B-ADV | B-GEN | B-RET | B-ORA |
|---|---|---|---|---|---|
| better | present | 0.9 | 4.6 | 14.0 | 32.6 |
| | absent | 2.4 | 2.5 | 3.2 | 1.4 |
| | Total | 3.3 | 7.0 | 17.2 | 34.0 |
| worse | present | 0.6 | 2.0 | 2.4 | 3.5 |
| | absent | 3.1 | 6.2 | 3.9 | 0.1 |
| | Total | 3.7 | 8.2 | 6.3 | 3.6 |
| # better rel. | | 11 | 13 | 34 | 39 |

(a) Percentage of times the answer appears in the top-$k$ retrieved paragraphs by DrQA. We use k=1 for our experiments as a single paragraph can already contain a large number of tokens.

(b) For T-REx, we report the percentage of time the model changes its output for the *better* or *worse* when the context is provided, grouped by the *presence* or *absence* of the answer in the provided context. B-RET and B-ORA scored higher than the context-free model on most relations.

Figure 1

entity but contains a distracting and semantically plausible answer $a'$. Table 4 shows some examples of adversarial contexts.

## 4. Results

The main results of our analysis are summarized in Table 2. It shows the mean precision at one (P@1) for the DrQA baseline and BERT-large on the LAMA probe enriched with different kinds of contextual information. Enriching cloze-style questions with relevant context dramatically improves the performance of BERT: B-ORA obtains ×7.4 improvement on Google-RE, ×1.9 on T-REx and ×3.5 on SQuAD with respect to using context-free questions (B). This clearly demonstrates BERT's ability to successfully exploit the provided context and act as a machine reader model. Remarkably, no fine-tuning is required to trigger such behaviour.

When we rely on TF-IDF retrieved context (B-RET), BERT still performs much better than in the no context setting. Overall, B-RET results are comparable with DrQA on Google-RE and SQuAD and much higher on T-REx. This is particularly surprising given that B-RET, unlike DrQA, did not receive any supervision for this task. Pairwise sign tests across relations show that the improvements for B-RET and B-ORA are indeed statistically significant (p-value below 1e-5).

Figure 1a shows the recall of the IR system, which demonstrates that the answer is not present in many of the retrieved contexts, though often the context is still related to the same topic. Table 1b reports a detailed analysis of whether the answer is present in retrieved contexts and how that affects the model's predictions. We observe that most of the gain of B-RET comes from cases in which the context contained the answer. However, there are also cases where the context does not explicitly mention the answer but BERT is still able

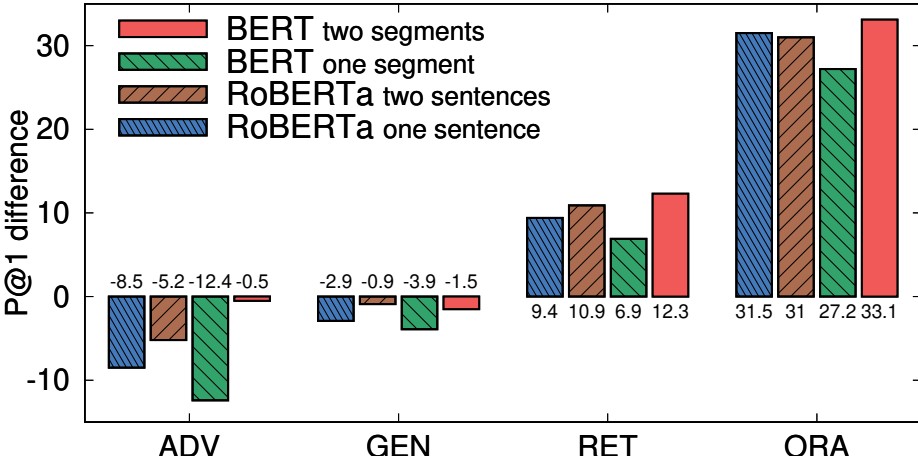

Figure 2: For each type of context considered, we report the change in P@1 relative to zero context, averaging results across relations. For each model we consider a concatenation of question and context as well as separating the two using separator tokens (BERT) or end of sentence tokens (RoBERTa). Separation dramatically improves both model's ability to ignore poor context and improves BERT's performance in the presence of good context.

to utilize the related context to help select the correct answer. Note that an extractive approach (such as DrQA) would have provided an incorrect answer (or no answer) for those cases.

### 4.1 Adversarial Robustness

The B-ADV column in Table 2 shows the LAMA $P$@1 results for BERT for adversarial contexts. BERT is very robust, dropping only 0.5 $P$@1 on average from the zero context baseline. However, as shown in Figure 2, this strong performance only occurs when the context and question are processed as two segments using BERT's separator tokens. Using only one segment (that is, simply concatenating the input query and the context) leads to a severe drop of 12.4 $P$@1 for BERT (a 40.7% relative drop in performance). We also observe a consistent improvement in performance from one segment to two for retrieved and oracle contexts.

One possible reason for this phenomenon resides in the Next Sentence Prediction (NSP) classifier of BERT, learned with self-supervision during pretraining by training the model to distinguish contiguous (*i.e.,* "next sentence" pairs) from randomly sampled blocks of text. We hypothesize that the MLM task might be influenced by the NSP's output. Thus, BERT might learn to not condition across segments for masked token prediction if the NSP score is low, thereby implicitly detecting irrelevant and noisy contexts. A result that seems in line with this hypothesis is that RoBERTa, which does not use NSP, is more vulnerable to adversarial contexts and the difference between one and two sentences (for RoBERTa separated by the EOS token) is much smaller.

To further investigate this hypothesis, we calculate the number of $(c, q)$ pairs classified by BERT as "next sentence" pairs in LAMA for the different context strategies. These results are shown in Table 3. We see that for B-RET, B-GEN and B-ORA, NSP classifications are

| (% Next Sentence) | B-ADV | B-GEN | B-RET | B-ORA |
|---|---|---|---|---|
| Google-RE | 10.4 | 95.1 | 88.9 | 98.4 |
| T-REx | 14.0 | 97.0 | 89.7 | 94.5 |
| SQuAD | 11.9 | 96.4 | 93.1 | 99.3 |

Table 3: Percentage of examples classified as 'next sentences' according to BERT's NSP classifier for the different context types. The low number of 'next sentence' classifications for B-ADV shows the model is able to recognize that adversarial contexts are unrelated and thus limit its influence on modeling the masked token in the query.

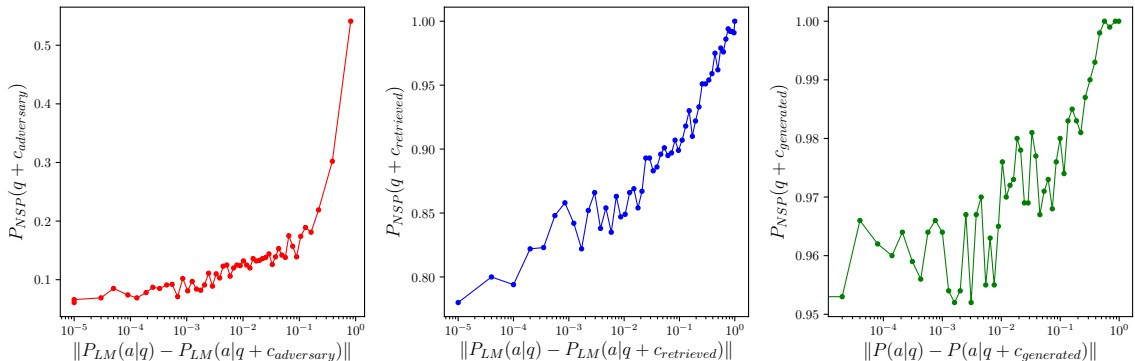

Figure 3: NSP Probabilities vs the change in LM probability upon appending contexts from the adversary (left), retrieval (mid), and generation (right) systems. At higher NSP probabilities, we see a higher larger increase to the probability mass placed on the correct answer in the presence of the context. That is, the more relevant that BERT thinks the context is, the more we see an increase to the likelihood of the true answer. This is exactly what we would want to see if we had hand-trained a relevance system ourselves, yet this instead emerges naturally from BERT's NSP pre-training loss.

high, suggesting BERT finds the segments to be contiguous, and hence useful to condition upon. However, for B-ADV, very few $(c, q)$ pairs are classified as "next sentences", suggesting BERT may condition on them less. Additional evidence for our NSP adversarial robustness hypothesis is given in Figure 3. Here we compute the absolute difference in probability that BERT places on the correct answer upon including context $||P_{LM}(a|q) - P_{LM}(a|q + c)||$, and plot it against NSP probability $P_{NSP}(q, c)$. We see that for adversarial, retrieved and generated contexts, increasing NSP probability is associated with greater change in true answer probability upon including context.[5]

---

5. Each context method has different NSP statistics, (*e.g.* the generated contexts have very high NSP probabilities on average) but the trend is consistent—higher NSP scores co-occur with greater changes in correct answer probability

## 4.2 Generated context

By generating context from a LM we aim at assessing the performance of a solution purely based on knowledge implicitly stored in the parameters of the underlying neural networks. Although the overall results of B-GEN are lower than the context-free baseline, some interesting insights emerge from our analysis. First, generated context improves performance for 13 relations and overall for 7% of the questions on T-REx (Table 1b). This demonstrates that autoregressive language models can generate relevant context and potentially serve as unsupervised IR systems. They do, of course, generate also irrelevant or factually wrong information. What is interesting is that BERT associates high NSP probabilities with generated contexts—for BERT, the generation is always a plausible continuation of the question. This inhibits the selective behaviour of BERT with respect to the context, and hurts performance when the generation is noisy, irrelevant or wrong.

Table 4 shows three examples for the generation of BERT-large for adversarial, generated, retrieved and oracle context-enriched questions.

## 5. Discussion

In this section we discuss some of our findings and their implications.

**Re-examining NSP**   The Next Sentence Prediction task has been extensively explored [Devlin et al., 2019, Liu et al., 2019, Yang et al., 2019c, Lan et al., 2019] with the apparent consensus that it is not helpful for downstream fine-tuning accuracy. Our findings, in contrast, suggest that it is important for robust exploitation of retrieved context for unsupervised tasks. Basing design decisions with a limited set of downstream uses when designing general purpose pre-trained models may well us lead to less flexible models. As a community, we should continue to strive for greater diversity in our criteria and possible use-cases for assessing such models [Talmor et al., 2019].

**Practical Takeaways**   Section 4 shows that BERT has a very different behaviour when inputs are processed with one or two segments. Practitioners should thus ensure that they thoroughly ablate segmentation options. The consistent improvement upon including retrieved context also suggests that it may be possible to get performance boosts in many other tasks by the trivial incorporation of retrieved documents, even when such documents are not strictly required for the task. We leave investigating this for future work.

**Comparison with DrQA**   We demonstrate that BERT with retrieved context and no fine-tuning performs on par with DRQA on the LAMA probe, but it is worth discussing this comparison further. Firstly, it is encouraging that an unsupervised system performs just as well as a system that requires significant supervision such as DRQA. We further note that LMs are *abstractive* models, whereas DRQA is *extractive*, confined to returning answers that are spans of retrieved context. However, it is worth stating that LAMA only requires single token answers. Generating an arbitrarly long sequence of contiguous tokens from bidirectional LMs like BERT and RoBERTa is not trivial, but extractive QA models handle such cases by considering spans of text of varying lengths. Finally, while we have chosen DRQA as our baseline to compare to recent work, there exist several

F. Petroni, P. Lewis, A. Piktus, T. Rocktäschel, Y. Wu, A. H. Miller, S. Riedel

| Query | Predictions |
|---|---|
| [P101] Allan Sandage works in the field of _____ . | engineering [-3.1] |
| ADV: q [SEP] According to Gould, classical Darwinism encompasses three essential core commitments: Agency, the unit of selection, which for Charles Darwin was the organism, upon which natural selection ... [0.0] | psychology [-2.8] economics [-3.4] anthropology [-3.5] |
| GEN: q [SEP] How many hours a week does he work? Does he get paid? How much does he get paid? How much does he get paid? He does not have a car. [1.0] | finance [-2.1] engineering [-3.4] advertising [-3.4] |
| RET: q [SEP] In 1922 John Charles Duncan published the first three variable stars ever detected in an external galaxy, variables 1, 2, and 3, in the Triangulum Galaxy (M33). These were followed up by Edwin ... [1.0] | **astronomy [-0.0]** physics [-5.5] observation [-7.3] |
| ORA: q [SEP] He currently works at the Institute of Astronomy in Cambridge; he was the Institute's first director.Educated at the University of Cambridge, in 1962 he published research with Olin Eggen and Allan ... [1.0] | **astronomy [-0.0]** physics [-4.0] galaxies [-5.5] |
| [P279] Interleukin 6 is a subclass of _____ . | proteins [-0.2] |
| ADV: q [SEP]First built in 1893 by Chinese residents of Nagasaki with the support of the Qing Dynasty government, the shrine was designed to serve as a place of worship and learning for the Chinese ... [0.0] | proteins [-0.2] **protein [-3.1]** DNA [-3.7] |
| GEN: q [SEP]Okay, let's get this out of the way. The Interleukin 6 (IL-6) is an interleukin-6 receptor (IL-6R) that plays a key role in the immune system. Intra-leukin-6 (IL-6R) is an interleukin-6 receptor (IL ... [1.0] | proteins [-0.6] receptors [-1.6] antibodies [-2.2] |
| RET: q [SEP]In particular, the increase in levels of IL-6 (interleukin 6), a myokine, can reach up to one hundred times that of resting levels. Depending on volume, intensity, and other training factors, the IL ... [1.0] | insulin [-1.9] IL [-2.1] proteins [-2.4] |
| ORA: q [SEP]It is a cardiac hypertrophic factor of 21.5 kDa and a protein member of the IL-6 cytokine family. This protein heterodimerizes with interleukin 6 signal transducer to form the type II oncostatin M ... [1.0] | proteins [-0.7] **protein [-1.5]** insulin [-2.4] |
| [P413] Giacomo Tedesco plays in _____ position . | center [-2.2] |
| ADV: q [SEP]On July 31, 2009 he was traded from the Tigers to the Seattle Mariners along with fellow pitcher Luke French for veteran pitcher Jarrod Washburn. On July 31, 2009 he was traded from the Tigers to ... [0.03] | center [-1.5] centre [-2.4] forward [-2.6] |
| GEN: q [SEP]How much does he play? He can play fullback, wing or centre. He can also play on the wing. Tedesco can also play in the halves. Tedesco can play in the halves. [1.0] | fullback [-1.4] centre [-2.1] wing [-3.6] |
| RET: q [SEP]Giovanni Tedesco has two brothers who are also football players, Salvatore (formerly of Perugia and Lucchese) and Giacomo, who is playing for Reggina. [1.0] | **midfielder [-1.2]** forward [-1.8] midfield [-2.3] |
| ORA: q [SEP]Giacomo Tedesco (born February 1, 1976 in Palermo) is a former Italian football (soccer) midfielder. Giacomo Tedesco (born February 1, 1976 in Palermo) is a former Italian football (soccer) midfielder ... [1.0] | **midfielder [-0.7]** forward [-2.2] defender [-2.4] |

Table 4: Examples of generation for BERT-large. We report the top three tokens predicted with the associated log probability (in square brackets) for adversarial (ADV), generated (GEN), retrieved (RET) and oracle (ORA) context-enriched questions. NSP probability (in square brackets) reported at the end of each statement.

more sophisticated supervised open-domain QA models that outperform it on a variety of open-domain QA tasks [Lee et al., 2019, Yang et al., 2019a, Guu et al., 2020].

**Unsupervised Question Answering** Our work is part of growing body of work that demonstrate that unsupervised question answering is not only possible, but beginning to reach and even outperform some standard supervised baselines. Radford et al. [2019] and Lewis et al. [2019b] demonstrate non-trivial performance on CoQA [Reddy et al., 2019] and SQuAD [Rajpurkar et al., 2016] respectively, and [Yadav et al., 2019] achieve SoTA results using an unsupervised method for multi-choice QA on ARC [Clark et al., 2018]. Taken together, these recent findings suggest that powerful and flexible unsupervised QA systems could soon be a reality, bringing with them many advantages including avoiding biases that often plague smaller datasets by incorporating knowledge from much larger corpora and greater abilities to combine and abstract pieces of information from different sources.

## 6. Conclusion

We demonstrated a simple technique to greatly improve factual unsupervised cloze QA by providing context documents as additional inputs. We used oracle documents to establish an upper bound to this improvement, and found that using off-the-shelf information retrieval is sufficient to achieve performance on par with the supervised DrQA system. We also investigated how brittle language models' factual predictions were to noisy and irrelevant context documents, and found that BERT, when featurized appropriately, is very robust. We provide evidence that this robustness stems from the Next Sentence Prediction pre-training task.

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
