# OpenReview forum: "How Context Affects Language Models' Factual Predictions"
_AKBC.ws/2020/Conference — AKBC 2020_

### Official Review · AnonReviewer3 · 2020-03-09
**Official Blind Review #3**

**Rating:** 7
**Confidence:** 4

**Review:**

This work analyses how factual predictions of a Masked Language Model (MLM) such as BERT and RoBERTa are influenced by adding extra context to a query. The paper examines a variety of ways of constructing this context, spanning over settings such as adversarially constructed, generated by a language model, retrieved by supervisedly trained systems, a TF-IDF retrieved baseline and an oracle. The paper finds that enriching a query with a good context can substantially improve performance in the LAMA probe, that analyses factual predictions. Additionally, the results demonstrate that there is considerable headroom for improvement in the retrieval side, evidenced by the results using an oracle retriever. Moreover, the paper shows the importance of BERT's Next Sentence Prediciton task, showing that it makes the model robust to adversarial appended contexts.

Overall, the paper is well written and the results are relevant to the community. As argued, completely relying on model's parameters for storing factual knowledge has a series of disadvantages compared to models able to retrieve relevant factual information from a corpus. This is especially relevant when this is done in an unsupervised manner, as it allows proper scaling. The experiments show clear evidence to support the claim that augmenting a query with a proper context greatly enhances performance on a factual knowledge probe. One strong point of this paper is the comparison with multiple strategies for generating contexts.

The paper claims to differ from previous work by considering a fully unsupervised setting. While it is true that no extra supervision is needed for the B-RET experiments, the exact same point holds for other work such as REALM (Guu et al, 2020), which the paper mentions. REALM is unsupervisedly pre-trained (including the retrieval portion). It would also be nice to see quantitative comparisons with the contexts retrieved by this model, though it's understandable that the authors don't report this, given how recent this work is and that it is not open-source at the time of writing.



Typos & other minor comments:
Section 2, Language Models and Probes: It's a bit of a stretch to call modells like T5 a "variant" of BERT.
Section 2, Open-Domain QA: "areas" - > area

---

> ### Author Response · Authors · 2020-04-09
> **Thanks!**
>
> We thank the reviewer for their feedback. REALM is indeed pre-trained in an unsupervised/self-supervised way, and we are looking forward to compare to it directly once we have access to the code. The reviewer mentions “other work such as REALM”--we would also love to compare against other work in this list, and if you do have references to more such work we love to learn about them.

---

### Official Review · AnonReviewer1 · 2020-03-26

**Rating:** 9
**Confidence:** 4

**Review:**

This paper shows unsupervised performance on LAMA when using various methods to obtain contexts.  It is very related to the recent REALM work (which was posted a few days before this submission); both show that transformers perform quite well when given related, retrieved context.  This paper does it in a fully unsupervised way, however, and includes some really interesting analysis.  I really liked all of the ways the models were probed, including using a generative model to provide context.  This at first seemed odd to me, but the authors provide a good justification for why this is an interesting probe in section 4.2.

The authors themselves noted the limitations of the work in the paper (e.g., single tokens vs. longer answers, mentioned on page 10), so there is little for me to mention as problematic.  My one minor quibble is with the "unsupervised question answering" section on page 10.  In the first sentence of section 6, the authors are careful to state that they are talking about "factual unsupervised cloze QA", but there is no such hedging in the unsupervised QA section just above.  There really is only evidence here for simple, factual, predicate-argument structure style questions, and using blanket, unqualified terms like "question answering" feels like over-claiming.

This review seems very short to me; mostly I write notes about things that aren't clear, or that could be improved, or aren't true.  I didn't really have anything to write about this paper.  The review is short because the paper is excellent, and I learned a lot from it.

---

> ### Author Response · Authors · 2020-04-09
> **Thanks!**
>
> We thank the reviewer for their feedback. We do agree our unsupervised QA section also only focuses on simple, factual, predicate-argument structure style questions. We will revise this in the final version of the paper.

---

### Official Review · AnonReviewer2 · 2020-03-27
**Great insights!**

**Rating:** 9
**Confidence:** 5

**Review:**

The paper explores how the performance of BERT and DrQA changes as a result of being applied to different text snippets. The paper compares retrieved snippets, generated snippets (NLG), adversarial snippets (answers to different questions), as well as an oracle (using the correct snippet of the extraction from Wikipedia).

This is a great paper that provides a lot of insights into how the quality of the underlying content affects the prediction quality.

I have very little to complain. I would have appreciated but some significance analysis on the results. I want to point out that TF-IDF is a very weak retrieval model, but I understand that this is not the focus of this paper.

---

> ### Author Response · Authors · 2020-04-09
> **Thanks!**
>
> We thank the reviewer for their feedback. Indeed, we agree that a significance analysis will help and aim to add it to the final version. We also agree that TF-IDF isn’t the strongest IR model (but surprisingly hard to beat for more complex ones in many cases) and are interested in improving our work along this dimension.

---

### Decision · Program_Chairs · 2020-04-30

**Decision:**

Accept

**Comment:**

This paper studies how factual predictions of a Masked Language Model (MLM) are influenced by appending additional context via various context construction methods. The work presents a set of interesting probes for the analysis, with good justification on the probe design. The paper is well written, clear, and provides good insights on understanding and improving MLM.